# Iron Oxide-Based Magneto-Optical Nanocomposites for In Vivo Biomedical Applications

**DOI:** 10.3390/biomedicines9030288

**Published:** 2021-03-12

**Authors:** Nisha Lamichhane, Shalini Sharma, Anita Kamra Verma, Indrajit Roy, Tapas Sen

**Affiliations:** 1Nano-Biomaterials Research Group, School of Natural Sciences, University of Central Lancashire, Preston PR1 2HE, UK; NLamichhane2@uclan.ac.uk; 2Department of Chemistry, University of Delhi, Delhi 110007, India; shalinidu1990@gmail.com (S.S.); singh.parul809@gmail.com (P.); indrajitroy11@gmail.com (I.R.); 3Nano Biotech Lab, Department of Zoology, Kirori Mal College, University of Delhi, Delhi 110007, India; akverma@kmc.du.ac.in

**Keywords:** iron oxide nanoparticles (IONPs), optical probes, magneto-optical, semiconductor, carbon quantum dots, up-conversion, plasmonic, theranostics, cancer, diagnostics, therapeutics

## Abstract

Iron oxide nanoparticles (IONPs) have played a pivotal role in the development of nanomedicine owing to their versatile functions at the nanoscale, which facilitates targeted delivery, high contrast imaging, and on-demand therapy. Some biomedical inadequacies of IONPs on their own, such as the poor resolution of IONP-based Magnetic Resonance Imaging (MRI), can be overcome by co-incorporating optical probes onto them, which can be either molecule- or nanoparticulate-based. Optical probe incorporated IONPs, together with two prominent non-ionizing radiation sources (i.e., magnetic field and light), enable a myriad of biomedical applications from early detection to targeted treatment of various diseases. In this context, many research articles are in the public domain on magneto-optical nanoparticles; discussed in detail are fabrication strategies for their application in the biomedical field; however, lacking is a comprehensive review on real-life applications in vivo, their toxicity, and the prospect of bench-to-bedside clinical studies. Therefore, in this review, we focused on selecting such important nanocomposites where IONPs become the magnetic component, conjugated with various types of optical probes; we clearly classified them into class 1 to class 6 categories and present only in vivo studies. In addition, we briefly discuss the potential toxicity of such nanocomposites and their respective challenges for clinical translations.

## 1. Introduction

Nanomaterials having the ability to simultaneously detect and cure various diseases is an emerging area of multidisciplinary research under nanobiotechnology due to their nanoscale dimensions [1]. Over the past decade, scientists have realized that multimodal nanoparticles can play an important role as theranostic agents in nanomedicine. Theranostics is a combination of diagnostic and therapeutic applications simultaneously using novel nanocomposites having multifunctional properties, thus, promising a ‘paradigm shift’ in the way medicine is practiced globally [2,3]. The ability to combine two or more complementary imaging modalities using carefully designed multimodal nanocomposites can help to generate comprehensive structural and functional diagnostic information [4]. The combination of therapeutic agents, especially those that can be externally activated by light or magnetic field and using them as a single nano-construct, facilitates not only diagnosis-driven targeted therapy but also real-time monitoring of therapeutic action at disease sites [5]. Substantial research has already been carried out on the applications of nanoparticles in medicine and led to the identification of certain positive trends that are likely to dictate the future roadmap on nanomedicine [6]. In particular, iron oxide nanoparticles (IONPs) have emerged as principal candidates due to their minimal toxicity in cellular systems and multimodal functionality such as superparamagnetism, localized heating under alternative magnetic fields (AMF), and magnetic resonance imaging (MRI) contrast enhancement [7,8,9]. In addition, IONPs have been utilized as photothermal agents for photothermal therapy (PTT) using light [10]. Similarly, the use of IONPs as a tracer for magnetic particle imaging (MPI) is being explored for producing three-dimensional images of IONPs distribution in tissues [11]. All these properties have been successfully utilized for unique diagnostic, targeting, and therapeutic purposes. For example, the ability to concentrate systemically administered iron-oxide nanoparticles into tumor sites using an external magnetic field is an excellent avenue for actively targeted delivery of minimally modified versions. MRI using T_2_ contrast enhancement of IONPs is widely applied in pre-clinical settings. AMF directed hyperthermia using these nanoparticles can render localized heating to the tumor site with minimum off-target effects as a potential therapeutic route. The ease of synthesis, surface modification, storage, handling, and transport of these nanoparticles are additional benefits that would pave the way for their successful clinical translation.

Similarly, organic dyes such as Rhodamine 6G (R6G) have been widely used as a fluorescence marker for monitoring the distribution of biological molecules into the target sites by using fluorescence spectroscopy [12]. However, there are drawbacks to using organic dyes, such as their low quantum yield and rapid photo-bleaching. This has resulted in the development of optical nanoparticles such as quantum dots with high quantum yield and photo-stability [12]. In addition, they can be excited at any wavelength, unlike optical dyes, and provide sharp and bright emission light of various colors by carefully tuning their sizes ranging from 2 nm to 10 nm. However, due to their toxicity, several other optical nanoparticles such as carbon/graphene dots, plasmonic and up-conversion nanophosphors have recently emerged. These optical probes are unique biological tags for monitoring their distribution in cellular systems by simple spectrophotometry as an imaging platform. In addition, many of them can also produce localized heat (hyperthermia) by activating them using a certain wavelength of laser light, hence can be utilized as therapeutic agents.

A combination of optical probes with IONPs can further enhance the repertoire of their biomedical applications. They also offer a myriad choice of nanomaterials, which can be selected for a specific biomedical application as diagnostic and therapeutic tools. Several review articles can be found that have chronicled separately either optical or magnetic nanostructures in terms of their synthesis, characterization, and biomedical applications. However, a comprehensive review of combining multimodal functionality in a single nano-construct and their application in real-life scenarios such as in vivo study, their toxicity, and status of clinical trials could immensely help nanomedicine researchers, and is currently lacking in the literature. Therefore, it is the first attempt to compile a review article with a focus on multimodal nanomaterials, which contain at least one optical probe in conjugation with IONPs (as magnetic component) for in vivo applications only. We have classified all possible optical probes in conjugation with IONPs into various classes and identified them as (i) molecule-based agents (e.g., dyes, photosensitizers), and (ii) nanomaterials. Nanomaterials include semiconductors (quantum dots), metals (plasmonic nanomaterials, metal nanoclusters), rare-earth-doped matrices (nanophosphors), dye-doped optically transparent matrices (e.g., dye/silica), and various forms of carbon such as graphene nanoparticles, nanotubes, and carbon dots. Finally, we have summarized their toxicity in in vivo biomedical applications and the current status of their clinical trial as magneto-optical nano-constructs. Figure 1 provides a schematic representation of various classes of magneto-optical nanomaterials described in this review and their multi-modality as theranostic agents for in vivo biomedical applications.

## 2. Various Classes of Magneto-Optical Nanocomposites

### 2.1. Class 1: Molecular Optical Probe-Iron Oxide Nanocomposites

Organic molecule-based optical probes are divided into two categories such as fluorophores as a diagnostic tool for optical imaging and photosensitizers as therapeutic tools for photodynamic therapy (PDT) and photothermal therapy (PTT). These molecular probes can be easily integrated on the surface of IONPs, either via direct conjugation or by incorporation/entrapment within a transparent micelle (lipid or polymeric) or silica shell upon the IONPs core. In each case, it is essential that the optical properties of the optical probes are not only preserved but also enhanced for some applications.

Optical imaging using organic molecules as fluorescent probes has been in practice for a long time now. Traditional organic fluorophores, such as rhodamine and fluorescein, have several drawbacks, such as limited optical range and a tendency to photo-bleach rapidly. Therefore, they have been primarily used in in vitro applications, such as labeling intracellular organelles. The advancement of new fluorophores and their properties, such as their emission in the near infra-red (NIR) range and resistance to photobleaching, aided by concurrent development in in vivo optical imaging techniques, has created huge interest in the nanomedicine research community. Currently, several such fluorophores are actively used not only in pre-clinical optical imaging investigations but also in human intraoperative procedures with some limitations, i.e., active targeting [13,14,15]. Therefore, their combination with IONPs further bolsters their versatility, which includes real-time monitoring of in vivo pharmacodynamics and bio-distribution of nanoparticles, in vivo microenvironment biosensing, analysis of excretion, and their potential long-term fate inside the body using multimodal imaging, etc. [16,17].

Pioneering work by Ralph Weissleder’s group paved the way for the development of iron-oxide nanoparticle-fluorophore conjugates for biomedical applications [16,17,18,19,20]. In their first report, they used a formulation of cross-linked iron oxide (CLIO) nanoparticles conjugated with the NIR dye indocyanine Cy5.5 to demonstrate dual-modality (combining MRI and optical imaging) for the visualization of axillary and brachial lymph nodes in vivo [16]. They further went on to demonstrate a series of exciting in vivo applications of the CLIO nanoparticles, which includes image-guided surgery of glioma (brain tumors) in rats [17], imaging of inflammation of pancreatic islets in diabetic mice [18], and atherosclerosis [19], and monitoring cardiomyocyte apoptosis in mice [20].

Several researchers have explored the possibility of formulating lipophilic IONPs within block-copolymer micelles, along with co-entrapping molecular fluorophores within the micellar shells. Vinod Labhasetwar’s group used Pluronic block copolymer-micelles to co-entrap the hydrophobic NIR fluorophores (SDB5700, SDA5177, SDA2826, SDA6825, and SDB5491) along with IONPs [21]. Using an external magnetic field for guidance, these nanoparticles were highly efficient in targeting tumor sites as compared to passive tumor targeting. Micelles of other amphiphilic co-polymers, such as poly(isobutylene-alt-maleic anhydride) and 1,2-Distearoyl-sn-glycero-3-phosphoethanol amine-N-amino(polyethylene glycol) (DSPE-PEG-NH_2_), were used by other groups to stabilize IONPs and to co-entrap NIR fluorophores for tumor-targeted multimodality imaging and bio-distribution analysis [22,23].

Glycol-chitosan nanoparticles entrapped several ferrimagnetic iron-oxide nanocubes (IONCs) with surface-conjugated NIR fluorophore Cy5.5 and a tumor-targeting peptide [24]. The clustering of IONCs has led to enhanced contrast enhancement in MRI studies. Combined near-infrared fluorescence (NIRF) imaging and MRI demonstrated the detection of tumors as small as 5 mm in diameter in mice (Figure 2). In another report, breast cancer screening in mice was executed using four different imaging modalities such as X-ray Computed Tomography (CT), MRI, and NIR optical and dual-energy mammography [25]. In this study, the authors demonstrated the importance of iron oxide-based ‘all-in-one nanoparticles’ when combined with NIR fluorophore DiR and silver sulfide nanoparticles (Ag_2_S-NP).

In another study using Cy5.5 and IONPs, MPI modality was combined with MRI and NIRF imaging to improve in vivo screening [26]. The monodispersed carboxylated IONPs were conjugated with -NH_2_ or -FMOC end- poly-ethylene-glycols (NH_2_-PEG-NH_2_ and NH_2_-PEG FMOC) via amide bonding and clustered nanocomposite formed after labeling with Cy5.5. Figure 3 shows the accumulation of nanoparticles in the liver and spleen through three different complementary imaging modalities using MPI, NIRF, and MRI. The accumulation of nanocomposites on these tissues was attributed to the dominance of Kupffer cells and splenic macrophages in the liver and spleen, respectively. Each of the modalities reveals specific information about the fate of nanocomposites, while their combination could provide sensitivity on their tracking for more precise in vivo screening.

Non-invasive and early detection of amyloid-β (Aβ) plaques in the brain is critical for the prevention of Alzheimer’s disease (AD). IONPs were combined with carbazole-based cyanine NIR fluorophores for successful delivery across the blood-brain barrier and ultrasensitive imaging of Aβ plaques in vivo in an APP/PS1 transgenic mice model [27]. In another report, Jansen and co-workers showed robust in vivo targeting of orthopedic implants aided by an externally applied magnetic field using fluorescent probes embedded core-shell magnetic (silica shell with IONP core) nanoparticles [28].

Another important class of molecule-based optical probes is photosensitizers (PSs), which upon excitation by visible-NIR light produces reactive oxygen species (ROS) through intermediate processes [29]. These macrocyclic molecules include chlorins, phthalocyanines, pyropheophorbides, photofrin, etc. [30,31]. Usually, PSs have higher excited state lifetimes, which allows them to undergo intersystem crossing into a triplet state, whereby they interact with either molecular oxygen to produce singlet oxygen or with nearby biomolecules to produce other ROS species. The selective destruction of targeted diseased cells and tissues by the produced ROS upon photoexcitation of PSs is known as photodynamic therapy (PDT). A PS molecule can be incorporated with IONPs using similar strategies as fluorophores, and their combination can lead to several exciting applications such as magnetically targeting along with MRI-guided PDT, combination therapy involving PDT, and Magnetic Hyperthermia Therapy (MHT) in the presence of an AMF.

In 2005, Raoul Kopelman’s group reported the co-incorporation of IONPs and the commercial photosensitizer “Photofrin” within polyacrylamide (PAA) nanoparticles containing surface conjugated F3 peptide [32]. Such magneto-optical nanocomposites used for T_2_ MRI contrasting efficiently imaged 9L glioma tumors orthotopically implanted in mouse brains. Subsequent laser light irradiation to the tumor led to a robust PDT effect and overall enhancement in the survival rate of tumor-bearing mice. This was followed by several other demonstrations of MRI-guided PDT. Sun et al. in 2009 used the biocompatible polysaccharide chitosan-coated formulation of IONPs combined with the porphyrin derivative as PS to study MRI and PDT in colon cancer in vivo [33]. Another group reported the use of functionalized Fe_3_O_4_ nanoparticles in combination with the PS chlorin e6 (Ce6, λ_ex_~645 nm) and PEG2K, and successfully demonstrated MRI and PDT effect in MGC-803 gastric cancer in vivo [34].

Combining IONPs with PS has led to a synergistic effect when applied in dual therapy involving MHT and PDT. Claire Wilhelm’s group co-encapsulated Fe_3_O_4_ nanoparticles and the PS Temoporphin (m-THPC: 5,10,15,20-Tetra(m-hydroxyphenyl) chlorin) within liposomes and injected in tumor-bearing mice for combination therapy [35]. This was carried out by exposure of the tumor area either with an AMF (30 mT, 111 kHz) or laser irradiation (100 mW), alone or in combination. Complete elimination of the tumor was observed in mice undergoing combination therapy, whereas mice undergoing MHT or PDT alone showed only a partial therapeutic effect. Another example of the combination of MHT and PDT in vivo was demonstrated using 10 nm superparamagnetic Fe_3_O_4_ nanoparticles combined with the photosensitizer pheophorbide a [36].

In addition to PDT, certain PSs (e.g., Indocyanine green) have been utilized as probes for PTT as they produce heat upon excitation by visible-NIR light. In combination with IONPs the PTT effect of such PSs can be accompanied with other modalities for therapeutic (PDT and MHT) and diagnostics (NIRF imaging, MRI, and thermal imaging) purposes. In a study involving ICG loaded IONPs, the nanocomposites were injected into 4T1 tumor-bearing mice, and a real-time temperature increase to 47.3 °C was observed under NIR laser irradiation (wavelength: 808 nm, power density: 2 W/cm^2^), for 5 min [37]. The thermal imaging data complemented the MRI and NIRF imaging verifying the accumulation of nanocomposites at the tumor site.

Recently, we have comprehensively reviewed class 1 molecular optical probes with IONPs for in vivo biomedical applications [38]. In the current review, we have covered selected examples on class 1 optical probes and explored fully classes 2 to 6 in combination with IONPs for in vivo biomedical applications.

### 2.2. Class 2: Semiconductor Nanoparticles-Iron Oxide Nanocomposites

Despite several recent advancements using molecular fluorescent probes for diagnostic purposes, they still suffer from photobleaching and quenching in in vivo studies, which limits their applications. Narrow absorption and broad emission spectra of organic fluorophore pose several practical challenges such as the requirement of specific light sources for excitation, the tendency for reabsorption/quenching by neighboring molecules, etc., for routine diagnostic applications. To overcome such limitations, semiconductor nanoparticles such as quantum dots have emerged as new-generation optical probes with broadband excitation and narrow emission spectra, as well as resistance to photobleaching [39]. Such quantum dots belonging to different classes of semiconductors, such as Groups II-VI, III-V, and IV (elements in the periodic table), and their combinations have been used for biomedical applications.

The above strategy was first applied by Shi et al. in 2009, utilizing quantum dots (QDs) immobilized Fe_3_O_4_ containing polystyrene nanospheres for simultaneous in vivo imaging and local MHT [40]. The selected QDs, with an emission wavelength of 800 nm having CdSeTe core and ZnS shell covalently functionalized with PEG, were immobilized on magnetic nanospheres by EDC: NHS (1-ethyl-3-(3-dimethylaminopropyl) carbodiimide/*N*-hydroxy succinimide) coupling and electrostatic adsorption. The nanoparticles, when administered to nude mice, provided a significant increase in fluorescence (excitation: 725 nm, emission: 790 nm) within one day after the injection. Additionally, the magnetic hysteresis showed the superparamagnetic properties of the nanoparticles, which upon application of AMF, increased the temperature to 52 °C within 30 min.

Similarly, another magneto-optical nanoparticle with a core-shell structure was synthesized by co-assembling CdSe-CdS core-shell quantum dots with Fe_3_O_4_ [41]. The resultant core-shell nanoparticles were further coated with a thin layer of silica offering uniform and tunable sizes, high magnetic plus fluorophore content loading, high colloidal stability, and biocompatibility. After functionalization with poly(vinylpyrrolidone) (PVP) ethylene glycol (EG), the functionalized nanoparticles were observed to be spherical in morphology with an average diameter of ~120 nm and were used in vivo multi-photon and MRI study. The nanoparticles were intravenously injected into mice bearing brain metastasis of a murine mammary carcinoma (MCaIV). Due to the high degree of aggregation of core MNPs in the resultant magneto-optical nanocomposite, the T_2_ relaxivity was 402.7 mM^−1 ^s^−1^, and was nearly 6.2 times larger than that of individual IONPs. A study using a combination of multiphoton and MRI revealed the accumulation of nanoparticles in the tumor region.

Cui and coworkers have established some pioneering research on the early identification of gastric cancer cells using magneto-optical nanoparticles of class 2 types [42,43]. In their early study in 2011, the group synthesized a fluorescent magnetic nanoparticle (FMNPs) composed of a silica-wrapped CdTe and IONPs of a size of around 50 nm and an attached anti-BRCAA1 antibody [42]. It was reported that BRCAA1 proteins are over-expressed in almost 65% of clinical specimens of gastric cancer tissues making it a suitable targeting molecule. In addition, the effect of different functional groups during the conjugation of anti-BRCAA1 antibodies on the surface of silica has been studied extensively. The conjugation efficiency of FMNPs with -COOH was found to be better than other functional groups. The nanoparticles were injected into nude mice with MGC-803 gastric cancer cells and studied for both optical imaging and MRI. The highest fluorescent signals were detected on the tumor site after 6 h of injection. The anti-BRCAA1-FMNPs nanoprobes preferentially accumulated in the tumor tissues, proven by MRI after 12 h of injection. Later, the same group utilized amino-modified silica-coated-FMNPs and labeled mesenchymal stem cells (MSCs) for magnetically targeted imaging and MHT in an in vivo gastric cancer model [43]. The MSCs labeled FMNPs showed fluorescent signals even after seven and 14 days of injection. The bio-distribution studies showed the nanoparticles accumulated only at the tumor site, as no specific fluorescent signal was detected in other major organs, indicating its’ specificity towards the tumor sites. To evaluate the effect of hyperthermia in the mice model, an AMF (Frequency: 63 kHz and Field: 7 kA/m) was applied for 4 min once every week for the period of one month. The inhibition of tumor growth (in volume) was clearly visible when MSC labeled FMNPs were injected in the mice following MHT. The group further analyzed the chemokine receptors’ expression levels in MSCs to investigate the mechanism of migration of MSC labeled nanoparticles on the gastric cancer cells in vivo. A concrete mechanism of MSC targeting was not possible; however, the study projected an increase in CXCL12-CXCR4 and CCL19-CCR7 by flow cytometry analysis.

Another study was carried on cyclic arginine-glycine-aspartate (cRGD)-conjugated magnetic-fluorescent (cRGD-MF) liposomes containing CdSe QDs as magneto-optical nanocomposite for targeted dual-modality imaging of bone metastasis from prostate cancer [44]. Hydrophilic IONPs along with hydrophobic CdSe QDs were encapsulated into liposomes with additional DSPE-PEG/DSPE-PEG-amines. cRGDyk (cyclic arginine–glycine–aspartic acid–tyrosine–lysine peptide) and the resultant nanocomposites were further conjugated with distal ends of DSPE-PEG-amines. The final nanocomposites were reported to be spherical in morphology with a diameter of ~120 nm. The fluorescence intensity of the cRGD-MF was lower than that of CdSe QDs alone. The nanocomposites were studied for in vivo MRI and fluorescence imaging using a mice model of bone metastasis from prostate cancer by intra-tibial injection of RM-1 cells. When using a dose containing ~15 mg of liposomes per kilogram body weight injected via tail vein, a strong MRI signal was achieved in the cRGD-MF liposomes compared with the unconjugated MF liposomes, as shown in Figure 4 (left panel). Similarly, in in vivo fluorescent imaging study, the cRGD-MF liposomes nanocomposites nicely accumulated on the tumor region, resulting in better signal enhancement compared to the unconjugated liposomes, as shown in Figure 4 (right panel).

### 2.3. Class 3: Carbon-Based Nanoparticles-Iron Oxide Nanocomposites

Carbon-based nanoparticles, including carbon nanotubes (CNTs), carbon quantum/nanodots (CQDs), and graphene oxides (GO), have been exploited in in vivo biomedical applications due to their optical properties. CNTs are cylindrical structured carbon molecules mainly classified as single-walled carbon nanotubes (SWCNT), double-walled carbon nanotubes (DWCNTs), and multi-walled carbon nanotubes (MWCNTs) based on the number of graphitic layers. CNTs in combination with IONPs have recently been applied for both diagnosis as enhanced hybrid MRI contrast agents and therapeutic by selective targeting, drug delivery, and MHT [45].

Doxorubicin-loaded SWCNTs tagged with IONPs followed by conjugation with Endoglin/CD105 antibodies have been used for active targeting and offering a theranostic approach with bioluminescence imaging (BLI) and MRI [46]. Delivering doxorubicin using IONPs-SWCNTs nanocomposites as drug delivery vehicles can reduce the toxicity and unnecessary side effects associated with nonselective bio-distribution. In their experiment, synthesized IONPs-SWCNTs were used for the conjugation with Endoglin/CD105 antibody through EDC: NHS coupling [46]. Doxorubicin binding was performed by physicochemical interaction with the magnetic CD105-conjugated SWCNTs. When injected intravenously on a tumor model of female Balb/c mice with 4T1-Luc2 cancerous cells, a significant decrease in primary tumor size was observed. Inhibition of tumor metastasis in the lungs following the injection of iron-tagged SWCNTs-Dox was also observed through BLI. T_2_ MRI contrasting also confirmed the enhancement in magnetic targeting towards the tumor site due to the progressive decrease in signal intensity in a time-dependent manner. Furthermore, to detect the treatment-induced changes in the tumor site, a diffusion-weighted MRI (DW-MRI) experiment was carried out where an apparent diffusion coefficient (ADC) value was directly related to the killing efficiency of tumor cells. It was observed that the magnetically targeted IONPs-SWCNTs produced significant changes in ADC values between 7–14 days after the injection, suggesting its efficiency in cancer treatment as a noninvasive imaging biomarker.

CQDs conjugated with IONPs have also been utilized for biomedical applications and are potential candidates for next-generation optical imaging owing to their excellent luminescence and photostability [47]. The release of free radicals from the magnetic CQDs accounts for most of its intrinsic cytotoxicity. Due to the release of free radicals on the bloodstream, bare IONPs can be unsafe for clinical translation, therefore, realizing the importance of surface coatings. γ-PGA is a natural anionic biopolymer that can be used as carbon and nitrogen precursors, simultaneously providing good water solubility and biocompatibility to inorganic nanocomposites [48]. The γ-PGA was used as a precursor and as a stabilizer to synthesize stable nitrogen-doped CQD-IONPs and labeled as C-Fe_3_O_4_ QDs with tri-modal in vivo bioimaging possibilities such as MRI, CT, and fluorescence (FL) imaging, as shown in Figure 5 [48]. The nanocomposites showed a transverse relaxation rate (r_2_) of 154.10 mm^−1^s^−1^ and an observable X-ray attenuation effect for CT imaging mode when injected in HeLa tumor-bearing nude mice. For evaluating the FL imaging, mice were positioned in a CRI Maestro in vivo fluorescence imaging system (excitation: 420 and 440 nm; emission: 520 and 570 nm). A strong FL signal was observed at the tumor site and bladder when excited at 420 nm (see Figure 5). In a bio-distribution study, the organs were excised after 24 h of injection, showing a significantly higher number of nanocomposites in the tumor site and in the kidney, as evaluated by inductively coupled plasma mass spectrometry (ICP-MS).

Other than the bioluminescent property of CQDs, another interesting property, “ROS scavenging from the cellular microenvironment” can be utilized to reduce the cytotoxicity of bare IONPs. Owing to these properties, CQDs doped IONPs were explored for multimodal imaging, ROS scavenging, and as a 3D printed polymeric nanocomposite for osteochondral tissue engineering [49]. The fluorescence property of the nanocomposites was evaluated in vitro and showed bright fluorescence restricted to the cytoplasm. The T_2_ MRI provided significantly higher contrast (relaxation rate: 118.3 mM^−1^ s^−1^), proving its usefulness for T_2_ contrast imaging and MRI-based cells/scaffold tracking. The T_2_ weighted MRI in vivo projected a high-intensity dark contrast on the kidney and bladder after 30 min of injection. Furthermore, to check the in vivo response and differentiation capability of the samples, the nanocomposites were cultured with MSCs for seven days in the presence and absence of an external magnetic field. The fluorescent and magnetoactive 3D printed composites were implanted subcutaneously in a rodent model. Formation of the osteoblast-like nested island was observed in the magnetically actuated samples providing evidence for the mechanical stimuli-responsive differentiation of cells. Furthermore, the downregulation of the *SOD* gene via PCR in vitro also suggested the role of CQD doped IONPs as ROS scavenging agents.

To enhance the PTT effect along with dual-modal MRI, Shi et al. in 2013 reported the use of a new class of nanoparticles (plasmonic, see detail in class 5) into IONPs-graphene oxide (GO) nanocomposites [50]. Polyethyleneimine (PEI, 1.2 kDa) was physically adsorbed onto GO-IONPs, followed by attaching seeds of gold nanoparticles using 1% hydrogen tetrachloroaurate (III) hydrate (HAuCl_4_). The resultant nanocomposites were further modified with folic acid (FA) conjugated lipoic acid (LA)-PEG. Under laser irradiation (808 nm, 1 W/cm^2^, 5 min), the heating efficiency of GO-IONP-Au (10 µg/mL) nanocomposites was double compared to GO-IONP alone. Tumors treated under laser irradiation (power density: 0.75 W/cm^2^) with GO-IONP-Au showed better ablation with negligible tumor cell growth.

Ma et al. in 2012 reported GO-IONP functionalized with PEG for in vivo MR imaging on 4T1 murine breast cancer model by magnetically targeted delivery of Doxorubicin and PTT guided by a magnetic field in 4T1 cell lines [51]. The T_2_-weighted MRI images of GO-IONP-PEG (200 µL of 2 mg/mL) injected intravenously in BALB/c mice bearing tumor showed a significant decrease (~67%) of signal intensity.

### 2.4. Class 4: Up-Conversion Nanoparticles-Iron Oxide Nanocomposites

Up-conversion nanoparticles (UCNPs) are a class of nanomaterials that show the unique, non-linear optical phenomenon of photon up-conversion, whereby excitation with long-wavelength (low energy) light leads to the emission of shorter wavelengths (higher energy) light. This is made possible when certain trivalent rare-earth cations (e.g., Ho^3+^, Er^3+^, Tm^3+^, and Yb^3+^) with ladder-like energy levels are doped in low concentrations within inorganic matrices with low phonon energy materials (e.g., NaYB_4_). One practical application of photon up-conversion is the feasibility of NIR excitation to generate emission in either the visible or NIR region, thus significantly improving the depth-profiling in optical imaging [52]. A growing interest in the use of up-conversion nanoparticles (UCNPs) in bioimaging is due to their up-conversion property, providing higher signal-to-noise ratios, deeper tissue penetration, low auto-florescence, and lower photodamage than conventional luminescent probes [53]. The application of UCNPs in bioimaging and their clinical trials are still restricted because of their low up-conversion luminescence (UCL) efficiency in vivo. To improve efficiency, UCNPs can be combined with IONPs, which is the main focus of this section of the review.

Recently, Fe_3_O_4_@Mn^2+^-doped NaYF_4_: Yb/Tm nanoparticles (NPs) were synthesized as promising imaging agents for NIR-to-NIR UCL and T_1_/T_2_-weighted MRI [53]. IONPs (Fe_3_O_4_) were used as seeds and covered with a NIR active shell of Mn^2+^-doped NaYF_4_: Yb/Er NPs using a facile hydrothermal method. To evaluate the NIR-to-NIR UCL imaging, a five-week-old nude mouse (male) was anesthetized and injected 10 mm beneath the abdomen region. The region of interest (ROI) analysis of the UCL signal (λ_em_ = 800 ± 12 nm) revealed a high signal-to-noise ratio (~26) between the abdomen region and the background. The ultra-low auto-fluorescence interferences with such a high signal could be attributed to the UCL of UCNPs featuring large anti-stokes shifts, in addition to the wavelength of the emission light that falls within the NIR window (700–1000 nm) for bioimaging. Furthermore, the T_1_-weighted MRI performed in vitro showed T_1_ relaxivity coefficient or relaxation rate (r_1_) as 4.7 mM^−1^ s^−1^, which is close to the relaxivity of commercial Gadolinium (Gd)-diethylenetriaminepentaacetic acid (4.82 mM^−1^ s^−1^).

Zhang et al. in 2012 synthesized ‘nanorattles’ consisting of SiO_2_-coated Fe_3_O_4_ and α-NaYF_4_:Yb/Er shells, fabricated through an ion-exchange process for targeted chemotherapy in vivo [54]. The nanoparticles could emit visible light as luminescence upon NIR irradiation, manipulated using an external magnetic field, and loaded with Doxorubicin. Upon NIR irradiation (wavelength: 978 nm), the visible spectrum with bands in two distinct regions was observed; one at around 510–570 nm (green region) and another at around 630–680 nm (red region). When intravenously injected in vivo and targeted magnetically, a significant increase in luminescence intensity was observed, demonstrating the accumulation of nanocarriers in tumors. The magnetic targeting of the Doxorubicin-loaded nanorattles (1 mg drug/kg of nanocomposite) resulted in a 96% reduction in tumor size in H22 tumor-bearing mice.

Another combination of UCNPs and IONPs was achieved by layer-by-layer (LBL) assembly of hexagonal NaYF_4_ (Yb:Er) UCNPs modified with polyacrylic acid (PAA) and iron oxides [55]. The dopamine-modified IONPs can be easily dispersed in water with excellent stability making them useful for the synthesis of UCNPs-IONPs nanocomposites. In their study, the UCNP-IONP was further modified with a layer of Au shell followed by Lipoic acid (LA) conjugated PEG. After intravenous injection of the modified UCNPs-IONPs (160 µL of suspension with suspension density of 2.5 mg/mL) in female nude mice bearing KB human epidermoid carcinoma tumors, the emission at 660 nm showed strong signals at liver and tumor sites suggesting higher uptake of nanocomposites. Later, Cheng and co-workers utilized the nanocomposite for an in vivo dual-modal optical/magnetic resonance imaging of 4T1 breast cancer tumor developed in Balb/c mice [56]. They showed that by placing a magnet nearby the tumor, the intravenously injected nanocomposites tended to migrate toward the tumor showing better contrast enhancement. The nanoparticle accumulation was around eight-fold higher than that without magnetic targeting. Furthermore, NIR laser irradiation (1 W/cm^2^) on tumors with an accumulation of nanocomposites under magnetic tumor-targeting showed an outstanding PTT efficacy with 100% tumor elimination.

A multi-modal in vivo imaging technique (UCL imaging/FL imaging/MRI) combined with drug delivery has been reported using Yb/Er-doped UCNPs (Y: Yb: Er = 78%: 20%: 2%) in combination with ultra-small IONPs encapsulated within an amphiphilic block copolymer PS16-b-PAA10 via a microemulsion method [57]. The study introduced a florescent dye (Squarine, SQ) into UCNPs-IONPs, and injecting in vivo intravenously; the UCL/FL imaging results are shown in Figure 6a. T_2_-weighted MRI of mice showed dark contrast in the liver (Figure 6b). The in vivo bio-distribution study of post-injected sacrificed mice showed a high accumulation of nanocomposites in the liver, spleen, and lungs, comparable with the FL and UCL imaging (Figure 6c). Averaged UCL and FL signals showed relatively high signal intensities in the liver compared to other organs (Figure 6d).

### 2.5. Class 5: Plasmonic Nanoparticles-Iron Oxide Nanocomposites

The optical features of plasmonic nanoparticles (PNPs) emanate from the property of localized surface plasmon resonance (LSPR), a resonant oscillation of surface electrons (plasmons) under visible light excitation. Nanoparticles of noble metals, such as gold and silver are well known for showing LSPR effects. Gold-based nanoparticles have been extensively used for optical bioimaging applications. Gold nanoparticles have a plasmonic signal in the visible region, whereas other nanostructures such as gold nanorods and nanoshells have tunable optical signal in the far red-NIR region [58]. Adding materials such as gold nanoshell protect the magnetic core from aggregation, oxidation, and corrosion. Likewise, gold nanoshells can also improve conductivity, optical properties, biocompatibility, surface functionalization, and chemical stability of IONPs [59,60]. In this context, interesting research was initiated nearly two decades ago to investigate the changes in magnetic properties of IONPs core by changing the thickness of gold shell [61]. Later in 2004, Lyon et al. reported the preparation of IONPs using co-precipitation of iron salts and the reduction of Au^3+^ to form a thin gold layer on its surface [62]. In 2007, Larson et al. reported that the magnetic-plasmonic formulation helped to tune plasmon bands to the NIR whilst maintaining the strong MRI contrast [63]. The combination of PNPs with IONPs has been utilized for the combined effect of MRI/optical imaging and PTT in MDA-MB-468 breast cancer cells with specificity. Even though the nanocomposites combined with PNPs and IONPs have been utilized for various in vitro applications, but there are very few in vivo studies. In this section, we have chronologically ordered and systematically evaluated combined PNPs-IONPs nanocomposites, specifically for in vivo biomedical applications.

Kim et al. in 2011, synthesized hybrid nanoparticles by combining IONPs with gold nanoparticles for in vivo applications using MRI and CT [64]. In their study, oleylamine stabilized Au-Fe_3_O_4_ nanoparticles were synthesized by mixing Au-oleylamine and Fe-oleate complex solutions. Then, the hybrid nanoparticles were coated with amphiphilic poly (DMA-r-mPEGMA-r -MA). The optical absorption spectrum of the polymer-coated hybrid Au-Fe_3_O_4_ displayed a red shift in the surface plasmon absorption band at 550 nm, which was presumably attributed to the junction effect of these two nanoparticles and the polymer coating layers. The micro-CT and 3T MRI images obtained in hepatoma-bearing mice showed a good contrast enhancement (~1.6-fold) after 1 h of injection of the nanoparticles. Furthermore, the T_2_ relaxivity coefficient (r_2_) obtained as 245 mM^−1^ s^−1^ was much larger than the commercial contrast agent, Resovist^®^, with an r_2_ of 150 mM^−1^ s^−1^. In 2012, Cai et al. reported a facile approach for the synthesis of iron oxide and gold nanocomposites (Fe_3_O_4_@Au NCPs) for MRI/CT dual in vivo imaging [65]. Fe_3_O_4_ was synthesized by a coprecipitation method and used as a core for subsequent electrostatic layer-by-layer assembly of PGA (poly(g-glutamic acid)) and PLL (poly(L-lysine)) to form PGA/PLL/PGA multilayers, followed by final assembly with dendrimer-entrapped gold NPs (Au DENPs). The Fe_3_O_4_@Au NCPs presented relatively high r_2_ relaxivity (71.55 mM^−1^s^−1^) compared to uncoated Fe_3_O_4_ exhibiting multifunctionalities for dual imaging modalities (MRI and CT imaging) of the liver and subcutaneous tissues in C57 mice.

To accomplish the aim of combining PTT and chemotherapy in a remote-control manner, Li et al. in 2014 reported Fe_3_O_4_@Au@mSiO_2_ nanocomposites using synthesized trisoctahedral core-shell Fe_3_O_4_@Au nanoparticles covered with a mesoporous silica shell and tested their efficiency in vivo on mice bearing Hela tumors [66]. The reactive oligonucleotides (referred to as double-stranded DNA (dsDNA)) were used as pore blockers for mesoporous silica shells that allowed the controlled release of Doxorubicin and labeled as a NIR responsive DNA-gated Fe_3_O_4_@Au@mSiO_2_ drug nanocarrier. By taking advantage of the magnetism due to IONPs, authors have managed to remotely trigger the drug release on the target site and monitored it with MRI imaging. Authors have reported a significant therapeutic effect on focusing the magnetic targeting of nanocarrier at the tumor site of mice bearing Hela tumors along with simultaneous irradiation of NIR laser light (wavelength: 808 nm, power density: 3 W/cm^2^), resulting in suppression of tumor growth due to the combination therapy.

Espinosa et al. in 2015 reported multicore IONPs (mostly Fe_2_O_3_) coated with Au shell and their application in vivo using dual therapies: MHT and PTT [67]. Maghemite (Fe_2_O_3_) multi-core IONPs were synthesized via polyol process and further functionalized with citrate anions for efficient seeding of Au followed by decorating with spiked or multi-branched gold shells. The magneto-plasmonic nanohybrids produced excellent heating efficiency with the temperature elevation of 6 °C by the application of AMF (frequency; 900 kHz and field: 25 mT) and NIR laser light (wavelength: 680 nm; power density: 0.3 W cm^−2^) simultaneously. This is equivalent to heating power and was calculated to have a SAR value of 634 ± 76 W g^−1^ of iron. The reason for such efficiency was explained earlier by Lartigue et al. in 2012, and it was due to the multi-core assembly potentiating thermal losses [68]. In the actual in vivo experiment, the nanocomposites were injected intratumorally on the subcutaneous tumor, and a temperature increase of almost 20 °C was observed within 2 min by the application of AMF and NIR laser light simultaneously. Moreover, the heating efficiency remained the same even after three days of injection, and consequently, tumor regression was observed to be measurable following five days of regular combination therapy.

Li et al. in 2015 reported the use of hyaluronic acid-modified Fe_3_O_4_@Au core/shell nanostars (Fe_3_O_4_@Au-HA NSs) for theranostic application using MRI, CT, Thermal imaging, and PTT [69]. Core/shell nanostars were fabricated by using Fe_3_O_4_@Ag nanoparticles as seeds for Au growth. The resultant core/shell nanostars (Fe_3_O_4_@Au-HA NSs) were further modified by sequential addition of polyethyleneimine (PEI) and hyaluronic acid (HA) for colloidal stability, biocompatibility, and for specific targeting towards CD44 receptor-overexpressing cancer cells, as shown in Figure 7i. HeLa tumor-bearing nude mice were used during the in vivo experiment, and nanocomposites were intratumorally injected and tested their efficiency by MRI and CT imaging. The tumor region darkened in a typical T_2_-weighted MRI study at 10 min post-injection with a significant difference in the signal intensity from 641 to 41, as shown in Figure 7ii (panel a). Additionally, the CT image enhancement was also observed upon post-injection, as shown in Figure 7ii (panel b). Upon NIR laser irradiation (wavelength: 915 nm; power density: 1.2 W/cm^2^) for the period of 90 secs, a rapid temperature increase was observed from 32.8 to 58.9 °C through the photothermal imaging.

In another study to reduce the number of nanoparticles being sequestered in the reticuloendothelial system (RES), Zhao et al. utilized the adipose-derived mesenchymal cells (AD-MSCs) with IO@AuNPs as a promising nanomaterial using a theranostic approach for injured livers and HCC [70]. IONPs of about 10 nm in diameter were coated with silica using the sol-gel method and seeded with gold. MRI and histological studies confirmed the IO@AuNPs loaded AD-MSCs preferentially accumulate on the diseased area. Later, the same group synthesized IO@AuNPs to label MSCs to test the feasibility of tracking carotid artery-injected superparamagnetic iron oxide nanoparticle (SPION)-Au via MRI and PA imaging for assessing the MSCs in glioma-bearing mice [71]. The localization of contrast agent labeled MSCs in neoplastic and ischemic lesions is a promising tool for the treatment of brain tumors such as Glioblastoma Multiforme (GBM). The addition of Au nanoparticles within the contrast labeled MSCs creates additional functionality for photoacoustic imaging (PAI), helping to map brain tumors. IO@AuNPs were stabilized by using PVP and amino-functionalized methyl-PEG. Green fluorescent protein (GFP) conjugated MSCs were labeled with IO@AuNPs at 4 μg/mL and injected via the internal carotid artery in six male athymic nude mice (NU/NU) bearing orthotopic U87 tumors. The in vivo T_2_-weighted MRI after injection of labeled MSCs showed progressive hypointensity of tumors over time. Similarly, an enhancement of the PAI signal was observed after the injection of labeled MSCs by comparing with unlabeled MSCs. The co-localization of GFP and iron from IONPs was confirmed via histological analysis even after 72 h of injection, indicating IO@AuNP labeled MSCs continue to carry their nanoparticle payloads, whereas the control MSCs did not show co-localization. Thus, the results suggest that a combination of PAI and MRI using IO@AuNPs is powerful for planning and real-time monitoring during stem cell-mediated therapy on brain tumors.

With an effort to study PTT and MRI on cancer cells initially in vitro, Shakeri-Zadeh and co-workers synthesized cysteamine-folic acid conjugated Fe_2_O_3_@Au nanoparticles [72]. The folic acid (FA) conjugated nanoparticles (hydrodynamic diameter within 30–70 nm; among less than a 3 nm gold layer) displayed a red shift in the surface plasmon band (absorption peak: 563 nm), making them well-suited NIR photothermal agents. Later, the same materials were utilized for PTT and MRI in vivo [73]. The bio-distribution of nanoparticles on CT26 colon tumor-bearing mice was monitored by MRI study, and therapeutic effect was monitored by irradiating NIR laser light (wavelength: 808 nm; power density: 1.4 W/cm^2^). The T_2_-weighted MRI of the tumor under an external magnetic field resulted in darker regions compared to the non-targeted area, confirming the accumulation of nanocomposites due to the application of an external magnetic field. Upon NIR laser irradiation on a magnetically targeted region of interest after intravenous and intratumoral injection, temperature rises of 12 °C and 16.7 °C were observed, respectively, showing an efficient PTT effect. Interestingly, the antitumor effect of nanocomposites by magnetically targeted localization followed by PTT with an intravenous injection was reported to be similar in efficiency to the intratumoral injection. However, as an intratumoral injection is not feasible for most tumor sites, therefore, intravenous injection is a preferable option—nanocomposite accumulation upon magnetic targeting and subsequent PTT by irradiating NIR laser light dramatically suppressed tumor growth.

Other than gold, silver is another plasmonic nanoparticle used in combination with IONPs for in vivo biomedical application. Ag@Fe_3_O_4_ with evenly enclosed carbon shells as multifunctional nanocomposites were used for theranostic application in vivo [74]. The nanocomposites were loaded with Doxorubicin and coated with PEG, followed by modification with folate. The resultant nanocomposites were reported to be around 140 nm in diameter, superparamagnetic in nature (saturation magnetization value = 102 emu g^−1^), drug loading efficiency of 17.5%, and a transverse relaxivity (r_2_) of 82.1 mM^−1^s^−1^ in vitro. In an in vivo MRI study on tumor-bearing mice, a high-intensity MRI signal at the tumor site was observed, which could be due to the deposition of nanoparticles through the EPR effect. Similarly, HeLa tumor-bearing mice were used for in vivo optical imaging study using NIR laser irradiation (wavelength: 808 nm; power density: 1.5 W/cm^2^). A temperature rises to 55 °C was reported after 5 min of NIR light irradiation. The fluorescence imaging after 24 h of injection showed the highest intensity due to the presence of nanocomposites in the tumor region, complementary with MRI. Furthermore, the tumor volume decreased significantly in the mice, indicating the high efficiency due to the nanocomposites’ formulation compared to Doxorubicin alone.

Curcio et al. designed core-shell nanohybrid with iron oxide and CuS for tri-modal cancer therapy with simultaneous MHT (SAR ~350 W/g), PDT and PTT (T ~ 46 °C, η ~ 42%) under AMF (frequency: 471 kHz, field: 18 mT) and laser light of wavelength 1064 nm [75]. CuS was chosen over other photothermal agents (i.e., gold or silver) owing to its NIR-II absorption with high PDT efficiency [76]. The authors used maghemite (α-Fe_2_O_3_) coated with NIR-II-absorbing plasmonic CuS shell (LSPR band centered at 1050 nm) as nanohybrid materials with a size of around 120 nm. Multicore iron oxide (IO) nanoflowers (SAR ~ 500 W/g) as magnetic cores are reported to be more efficient for localized heating over single domain nanocrystals. Authors have explained that nanoflowers do not form dipole-dipole chains, which inhibit their aggregation [77]. Nanohybrid induced ROS even in the absence of laser and stimulate ROS production in the presence of laser, indicating their potential use as PDT agents. In an in vivo study using tumor-bearing mice, nanohybrids showed an excellent PAI (wavelength = 905 nm). Furthermore, even at a low concentration of nanohybrids (0.6 mM Cu, 0.1 mM Fe), the temperature of tumors reached 53 °C under laser light irradiation for only 10 min whereas the similar temperature rise observed under MHT required nearly a thousand folds higher dose (650 mM Cu, 100 mM Fe), highlighting the superiority of the PTT effect of the nanocomposites over MHT. Another study showed that an IONP core and CuS shell nanocomposite with Doxorubicin could be a platform for tumor chemotherapy, PTT, and PDT [78]. Doxorubicin release could be triggered by the presence of gelatinase, showing enzyme responsive release in MCF-7 cells along with a synergistic cytotoxic response by laser irradiation (wavelength: 980 nm), leading to enhanced intracellular ROS generation compared to Doxorubicin alone. The temperature increased to 42 °C within 2 min of laser irradiation. Table 1 summarizes the combination of IONPs and class 5 nanocomposites for in vivo biomedical application.

### 2.6. Class 6: Other Optical-Iron Oxide Nanocomposites

In this section, we present all other optical probes which do not belong to the above five classes and have been utilized for in vivo studies. Surface modification of inorganic nanoparticles is important for biomedical applications as it improves biocompatibility, colloidal stability, and hydrophilicity. In this context, covalent bonding of polymers on inorganic nanoparticles are usually performed through two major approaches (i) ‘grafting to’ and (ii) ‘grafting from’. The ‘grafting to’ method allows functionalization through chemical reaction on the surface of nanoparticles with low-density polymers. Whereas the ‘grafting from’ method uses an initiator surface where layers of high-density polymers are grafted, leading to high graft density nanocomposites. In 2011, Li et al. used a specific polymerization technique called reversible addition-fragmentation chain transfer (RAFT), where a florescent carbazole agent was functionalized on magnetic silica nanoparticles as a chain transfer agent and N-isopropylacrylamide (NIPAM) as the monomer [81]. The resultant nanocomposite displayed effective negative contrast MRI. The suggestion on synthesizing a fluorescent polymer for efficient MRI was adopted by Yan et al. in 2014 to synthesize polymeric micelles with a fluorescent polymeric shell and core IONPs with a hydrodynamic diameter of 146 nm [82]. A carbazole containing monomer, 9-(4-vinylbenzyl)-9H-carbazole, was polymerized with 2,2,3,4,4,4-Hexafluorobutyl methacrylate (HFMA) by free radical polymerization to produce an amphiphilic poly(HFMA-co-VBK)-g-PEG copolymers that acted as a shell for encapsulating Fe_3_O_4_ magnetic nanoparticles. The micelles exhibited paramagnetic properties (Ms: 9.61 emg/g) and a transverse relaxivity rate of 157.44 mM^−1^ s^−1^. The in vivo MRI study in an SD mouse model showed high contrast region on the liver and spleen, suggesting nanocomposites uptake by the reticuloendothelial system (RES). The ex vivo images obtained by 2-photon (690–1040 nm) laser confocal microscopy (CLSM) demonstrated unique blue fluorescence in the liver showing distinctive characteristics of the carbazole containing Fe_3_O_4_-encapsulated polymeric micelles. Thus, the nanocomposites showed their potential for clinical use as MRI and optical imaging agents.

Polymers such as PDA (Polydopamine), a dopamine-derived synthetic polymer, exhibited a promising PTT effect when combined with IONPs in cancer theranostics. Recently, Li et al. constructed a nanocomposite of Fe_3_O_4_ coated with a PDA shell, which provided excellent stability with improved PTT [83]. The nanocomposites showed temperature rise to 57 °C upon laser light irradiation (wavelength: 808 nm laser, Power density: 1 W/cm^2^). The transverse relaxivity (r_2_) was reported to be 337.8 mM^−1^ s^−1^, which was comparatively higher than the commercial MRI contrast agents such as Resovist (143 mM^−1^ s^−1^). The nanocomposites were intravenously injected in the 4T1 bearing mice for evaluation of therapeutic effect and MRI study. Following laser irradiation, the tumor size decreased dramatically without any recurrence within 15 days of study. After 24 h, the T_2_-weighted MRI signal increased significantly on the tumor site.

Some of the lanthanide-based complexes can also act as extremely useful optical probes due to their excellent luminescence properties, such as large stokes shifts, high quantum yield, narrow emission bandwidth, longer lifetimes, and higher photostability compared to other optical probes [84]. Wang et al. in 2015 reported a novel nanocomposite by integrating IONPs (e.g., Fe_3_O_4_), rare earth fluorescent element Europium and small iodine as a contrast agent for multimodal imaging [84]. The nanocomposites possessed good paramagnetic properties with a maximum saturation magnetization of 2.16 emu g^−1^, transverse relaxivity rate (r_2_) of 260 mM^−1^S^−1^. The T_2_-weighted MRI on Sprague–Dawley (SD) rats showed relatively higher darkening on the liver and spleen compared to the kidney and muscles, indicating uptake of nanocomposites by the macrophages of RES. When subjected to in vivo optical imaging by 2-photon confocal scanning laser microscopy (CLSM), vivid fluorescent red dots were observed from the liver and spleen, complying with the presence of Eu^3+^ ions in the nanocomposite, exhibiting characteristic red luminescence. The CT images, when studied in vitro, showed increased intensity with an increase in iodine payload on the nanocomposites, thus showing the possibility of being tri-functional contrast agents having great clinical potential in CT, MRI, and optical imaging. Self-assembled magnetic luminescent hybrid micelles containing rare earth Eu have also been reported in in vivo experiments for dual-modality, i.e., MRI and optical imaging [85]. The resultant nanoparticles self-assembled to form magnetic and luminescent hybrid micelles and exhibited spherical morphology, paramagnetic properties (maximum saturation magnetization of 7.05 emu g^−1^), and a high transverse relaxivity of 340 mM^−1^ s^−1^. When applied in in vivo MRI study, the nanocomposites showed excellent contrast of the liver and spleen. Fluorescence spectra showed characteristic emission peaks from the rare earth Eu at 616 nm and vivid red fluorescence by 2-photon confocal laser scanning microscopy (CLSM).

## 3. Toxicity and Challenges for Potential Clinical Trials

Concurrent with the development of novel biomedical applications of nanomaterials, serious questions have been raised about the possible pitfalls of their use, particularly for in vivo administration due to unknown cytotoxicity [86]. This had led to some extensive in vivo toxicological analyses of such nanoparticles, especially their chemical composition, sizes, shapes, chemical stability, surface properties, route of administration, tendency for biodegradation, and their biocompatibility. Considering chemical composition perspectives, it has been generally agreed that most inorganic nanoparticles, such as iron oxide, silica, carbon, and gold, are mostly non-toxic [87,88]. However, detailed investigations have revealed that owing to the degradation, and poor elimination of ‘innocuous’ nanoparticles, i.e., iron oxide, several potential biohazards, such as hemolysis, anaphylaxis, sepsis, off-target organ toxicity, etc., may occur [89,90,91].

The majority of the nanocomposites described in this review article brings an additional toxicological perspective due to their optical components with magnetic IONPs, therefore, requiring a new set of investigations. The obvious physical aspect of IONP modification will be the size and shape. Generally, IONPs alone are usually considered spherical, ultra-small in diameter, and monodisperse; however, their modification with optical probes, particularly nanoparticulate forms, can lead to shape deformation, increase in size, alteration of surface charge, and polydispersity in suspension. In connection with the alteration in surface charge and shell material of IONPs based core/shell nanoparticles, a significant change due to protein adsorption (formation of protein corona) is observed, thus, influencing how cells recognize them [92]. Such physical transformation comes with the risk of capture by the reticuloendothelial system (RES), altered pharmacokinetics/pharmacodynamics, non-specificity, long-term persistence in the body, modulating uptake pathways, etc. Moreover, their combination with heavy-metal containing optical probes, such as II–VI quantum dots, can lead to enhanced toxicity. Another key factor while considering IONPs linked to optical probes is the potential for light-induced toxicity if these nanocomposites end up in superficial off-target organs such as the skin and eye [93].

Uptake of IONPs in the cellular system and the mechanism of toxicity are well documented from a biological perspective [94] whereas the combination of optical probes with IONPs is pre-mature and requires further study. Based on the current literature, with well-known mechanisms of toxicity [95], we have summarized the possible biological pathways of IONPs-based nanocomposites induced cellular toxicity, and an extension to opto-magnetic nanoparticles has been proposed in this article. Figure 8 and Figure 9 represent schematic diagrams of possible pathways for cellular toxicity.

Biomolecules that are charged or polar are unable to cross the hydrophobic plasma membrane; hence they can be internalized by active transport called endocytosis. Size-dependent cellular uptake of iron oxide-based nanocomposites occurs through different mechanisms such as passive diffusion, receptor-mediated endocytosis, clathrin-mediated endocytosis, caveolin mediated endocytosis, etc., result in a myriad of intracellular responses causing major cellular stress (see Figure 8). Large molecules (in the range of 250 nm) are taken-up by large vesicles, and pinocytosis encompasses uptake through small vesicles in the range of a few to hundreds of nanometers [96]. Pinocytosis can be subclassified into caveolin-mediated endocytosis, clathrin-mediated endocytosis, caveolae- and clathrin-independent endocytosis, and macro-pinocytosis [97]. Furthermore, specific molecules may be internalized in the cells through specific receptors, such as cholesterol via low-density lipoprotein receptor and iron (Fe^3+^) via transferrin receptor facilitated by Clathrin-mediated endocytosis. Once inside the cell, clathrin coatings on the exterior of the vesicles are ejected prior to fusion with early endosomes. The cargo (nanoparticulates or nanocomposites) within early endosomes may ultimately reach lysosomes via the endo-lysosomal pathway. In caveolin-mediated endocytosis, caveolae, internalizing molecules in the range of 50–80 nm, are composed of the membrane protein caveolin-1 that gives them a flask-shaped structure. Caveolae exist in epithelial cells, endothelial, skeletal muscles, adipocytes, and fibroblasts cells [98]. Cell signaling and regulation of membrane proteins, lipids, and fatty acids are controlled by Caveolae-dependent endocytosis. After caveolae are separated from the plasma membrane, they bind with a compartment of cell called caveosomes, which maintain a neutral pH. Caveosomes circumvent lysosomes, thereby protecting the contents from lysosomal degradation by hydrolytic enzymes. Therefore, pathogens, including bacteria and viruses, enter through this route to avert degradation. This pathway is exploited in nanomedicine as the cargo that is internalized by a caveolin-dependent mechanism evading lysosomes [97]. Other mechanisms of endocytosis such as clathrin- and caveolin-independent endocytosis occurs in cells when the cell surface lacks clathrin and caveolin. The dissolution of lysosomes increases the iron ions (Fe^2+^) release into lumen that further increases calcium levels leading to a partial loss of mitochondrial membrane potential (MMP) and eventually causing ATP depletion; hence Cytochrome c (Cyt c) release responsible for gene alteration. The elevated ER stress helps in the activation of effector caspases that trigger p53 dependent apoptosis. The accumulation of free iron ions (Fe^3+^) in cell lumen initiates oxidative stress and the Fenton reaction, which are reported to be two main reasons for cellular toxicity [94]. The co-existence of iron ions (Fe^+2^) from IONPs and H_2_O_2_ in the cellular system initiates the Fenton reaction (Fe^2+^ + H_2_O_2_ → Fe^3+^ + OH + OH^−^) and produces hydroxyl radicals (OH), a powerful oxidant creating additional oxidative stress on the tumor cells.

Excess iron ions can play a vital role in initiating the ferroptosis pathway. The Fe^3+^ ions in circulation, along with the additional iron ions (Fe^2+^) released from nanocomposites, may enter into the cells through the transferrin receptor (see Figure 9). The Fe^3+^ ions get reduced to Fe^2+^ under the catalytic effect of STEAP3 (six-transmembrane epithelial antigen of prostate 3), i.e., iron oxide reductase [99]. The Fe^2+^ ions accumulate into the labile iron pool (LIP) with the help of a divalent metal transporter 1 (DMT1). This may be due to the electron transfer ability and high solubility of Fe^2+^. LIP holds iron from the degradation of ferritin (ferritinophagy) and endosomal uptake of circulated iron ions. Increasing LIP formation may trigger the Fenton reaction and enhanced ROS level followed by ferroptotic cell death. Furthermore, System Xc^−^ mediates the uptake of glutamate and cystine that are constituents of glutathione (GSH), an important detoxifying molecule in the cells. However, the inhibition of System Xc^−^ impairs the intake of cystine following lipid peroxidation [100]. The lipoxygenases (LOXs) mediate the oxidation of PE-PUFAs to PE-PUFAs-OOH, eventually leading to ferroptosis. GPX4 plays a role as a protector to transfer PE-PUFAs-OOH to PE-OH and IREB2 as an iron-responsive binding protein 2. ROS-induced autophagy is another pathway that involves the destruction of (A) cytosolic proteins and organelles, (B) AMPK, (C) PI-3K and the formation of phagosome and autolysosomes. IONPs enter cells via endocytosis and stimulate the formation of phagophores, which support cargo sequestration. After completion, phagophores grow as autophagosomes and fuse with lysosomes to eventually form autolysosomes that will discharge their contents into the lumen. Iron ions either induce inhibition of autophagy through alteration in Akt/mTOR pathway or lysosomal impairment to trigger autophagy.

Recently Huo et al. have reported possible pathways for cellular toxicity by optical nanoparticles induced PTT and PDT [101]. Irradiation of NIR laser light (wavelength: 1064 nm) induces cell necrosis during PTT, whereas in combination therapies (PTT + PDT), both apoptosis and necrosis pathways are responsible for cellar toxicity. In addition, an anticancer effect can be induced by the endoplasmic reticulum and mitochondria due to the combination of PTT-PDT simultaneously, responsible for immunogenic cell death (see Figure 10).

Translation of opto-magnetic nanocomposites presented in the current review in theranostic applications raise new challenges due to difficulty in controlling their physical characteristics such as size, shape, and uniform chemical composition. Variation of such physical factors leads to altered toxicological profiles in vivo, owing to uncertainty in pharmacodynamics, biodistribution, clearance, etc. It is well known that through exquisite control of surface coating of the nanocomposites, their physical properties along with toxicological aspects can be modulated.

Denis et al. observed that coating a dense dextran layer around the IONPs reduces their immunogenicity [18]. Yen et al. reported ultra-small NIR fluorophore-IONP hybrid nanoparticles showing minimal cytotoxicity [22]. Using histological analysis, Chen et al. probed the potential organ toxicity of a targeted fluorophore-conjugated IONP formulation following intravenous injection in mice bearing pancreatic tumor. Other than mild inflammation, no significant toxicity in normal organs was observed [23].

The importance of proper surface modification is extremely significant in the pursuit of a uniform formulation involving two different nanoparticle types. Wang et al. did not observe any general damage to normal organs following in vivo administration of magneto-fluorescent nanoparticles (silica-wrapped CdTe and IONPs, diameter ~50 nm) in a murine model of gastric cancer [42]. In a separate example, Wang et al. showed CdSe QDs and IONPs within liposomal coating makes the resulting nanocomposites non-toxic [44].

From a clinical trial perspective, significant efforts have been made for the translation of pre-clinical laboratory work to real-life clinical application and their commercialization of IONPs due to low toxicity. For example, Ferumoxytol was first approved by FDA as an iron supplement to treat anemia (iron deficiency) in adult patients with chronic kidney diseases. This has now been widely utilized as “off label” nanosuspension for vascular imaging, detection of stem cell transplantation, and cancer imaging purposes [102]. Recently, we have reported commercially available IONPs with their trade names and current stages of clinical application on specific diseases [103]. During the literature search, we found that some of the IONPs showed better results, whereas others were withdrawn from pre-clinical and clinical studies due to their toxicity.

NANOM was the first-in-man (FIM) trial to assess the safety and efficacy of silica-gold (Nano group) and silica-iron-gold (Ferro group) nanoparticles containing IONPs completed recently for plaque management in atherosclerosis (ClinicalTrials.gov, Identifier: NCT01270139) [104]. A total of 180 patients received either Nano group (*n =* 60) as a patch in transplanted stem cells onto the cardiac artery or, Ferro group (*n =* 60) with CD68 targeted micro-bubbles and stem cells using a magnetic navigation system (magnetic field of 0.08 Tesla) verses stent implantation (*n =* 60). The nanocomposites were activated with NIR laser (wavelength: 821 nm, power density: 35–44 W/cm^2^) for 7 min of exposure. The Nano-group was observed to be safe in clinical practice associated with significant regression of coronary atherosclerosis as evaluated by the total atheroma volume (TAV) at the end of 12 months. While the Nano-group was considered safe, the Ferro-group, in comparison, showed toxic effects with a major adverse cardiovascular event (MACE).

Signs of nanotoxicity of the iron-bearing nanocomposites could be due to their underlying cytotoxic effects, such as the generation of reactive oxygen species [105,106], impairment of mitochondrial function, gene alteration, inflammation, chromosome condensation, and DNA damage [107]. Mahmoudi et al. in 2011 showed that the surface charge of the IONPs dictates the alteration of a number of gene expression patterns [106]. The presence of -COOH groups on IONPs was responsible for altering genes associated with cell proliferative responses. However, such gene alteration was subject to a specific cell type, indicating that nanoparticles with specific functional groups can play a significant role in defining the suitable pathways of nanoparticle toxicity. Additionally, the concentration of IONPs in the nanocomposite needs to be considered as a higher concentration of IONPs can cause irreversible DNA damage. pH can play an important role in controlling the nature (shapes, sizes, and zeta potential) of IONPs cores during their fabrication [108]. However, in the context of this review article on in vivo biomedical applications, the pH of a cellular system can also influence the formation of iron ions (Fe^2+^) which are responsible for Fenton reaction and additional oxidative stress, especially in tumor sites where the pH is mostly acidic. The variation of shapes and sizes of IONPs core with optical probes may have an influence on the cellular diffusion hence the formation of iron ions (Fe^2+^) in the tumor sites.

## 4. Conclusions and Future Perspectives

Detection of diseases and their cure using nanomedicine is an emerging trend in the area of nanotechnology, nano-biotechnology, and medicine. In this context, we have used one of the most applicable magnetic nanoparticles based on iron oxides (IONPs) and collected up-to-date literature information where IONPs have been conjugated with various optical probes. In this review, we have systematically classified all available optical probes from molecule-based probes (Class 1) to various optical nanoparticles (Classes 2 to 6) having fluorescence, plasmonic and up-conversion properties for especially focusing only on in vivo studies for biomedical applications. Classical molecule-based optical probes labeled as class 1 are well-known for bio-imaging, and several such probes are commercially available in spite of their limitation due to photobleaching and low quantum yield. Attaching them to IONPs overcomes such limitations due to their bimodal imaging (MRI and fluorescence) and multimodal therapeutic (PTT, PDT, and MHT) effects. Similarly, optical nanoparticle probes conjugated with IONPs create a range of opto-magnetic nanocomposites and their application in in vivo biomedical studies due to their multimodal imaging ability (MRI, CT, thermal, and fluorescence imaging) and multimodal therapeutic effects (in MHT, PTT, and PDT). Due to the toxicity of most of the semiconductor quantum dot nanoparticles (class 2), we have reviewed relatively non-toxic carbon-based nanoparticles (class 3) for in vivo biomedical applications. Among class 3 optical probes, CQD are reported to be promising nanomaterials due to their ultra-small sizes and fluorescence properties. The latest developments on up-conversion nanoparticles in conjugation with IONPs showed interesting fluorescence images where excitation was carried out using NIR laser light, having the advantage of high human tissue penetration depths. Similarly, we have identified the importance of plasmonic nanoparticles conjugated with IONPs for efficient multimodal diagnostic platforms such as CT and thermal imaging along with MRI. However, due to the toxicity issues associated with several optical probes, clinical applications and commercialization steps are a huge challenge; therefore, very little progress has been made in translating in vivo research to clinical trials. Finally, due to the multimodal therapeutic and diagnostic properties of opto-magnetic nanocomposites, they have huge potential as future nanomedicines, especially by selecting already commercialized IONPs in conjugation with clinically approved commercial optical dyes such as Indocyanine Green (ICG), etc. This review will immensely contribute to nano-biotechnology and nanomedicine communities due to access to up-to-date information of various classes of optical probes in conjugation with IONPs in the latest developments in in vivo studies at the pre-clinical stage.

## Figures and Tables

**Figure 1 biomedicines-09-00288-f001:**
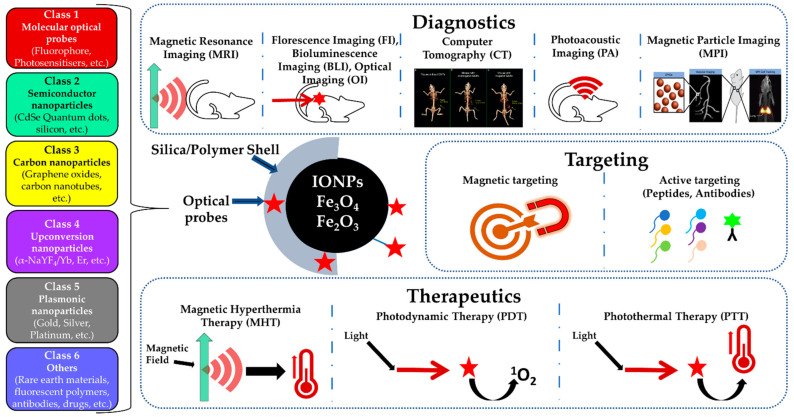
Schematic representation depicting iron oxide-optical probe conjugated nanocomposites for various in vivo biomedical applications. Different colors in the left represent different classes of optical probes. Green arrow indicates the magnetic field, red waves indicate the radiation, red stars indicate optical probe and thermometer symbols indicate temperature elevation due to heating.

**Figure 2 biomedicines-09-00288-f002:**
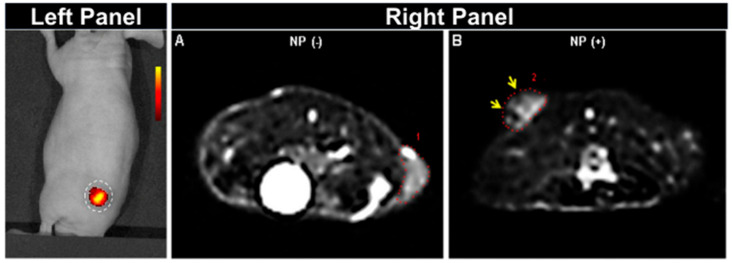
Different imaging modalities on athymic mice demonstrating tumor accumulation of nanocomposites after 24 h of injection. (Left Panel): In vivo NIRF imaging showing fluorescent tumor site, (Right Panel): 3T MRI. Red dotted lines 1 and 2 show the edge of the tumor before (**A**) and after (**B**) nanocomposites administration. Reprinted with permission from ref. [24]. Open Access Creative common.

**Figure 3 biomedicines-09-00288-f003:**
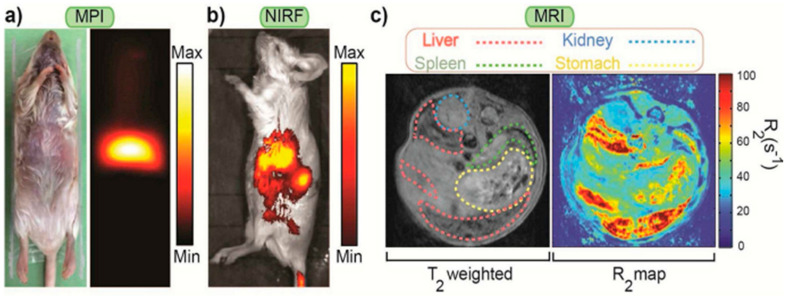
Complementary imaging modalities (**a**) colorized magnetic particle imaging (MPI) (**b**) near-infrared fluorescence (NIRF) and (**c**) MRI T_2_ weighted and colorized R_2_ images 72 h after injection of nanoparticles (NPs) functionalized with NH_2_-PEG-FMOC (100 μL, 1 mg Fe/mL). Reprinted with permission from ref. [26]. Copyright 2015 Elsevier.

**Figure 4 biomedicines-09-00288-f004:**
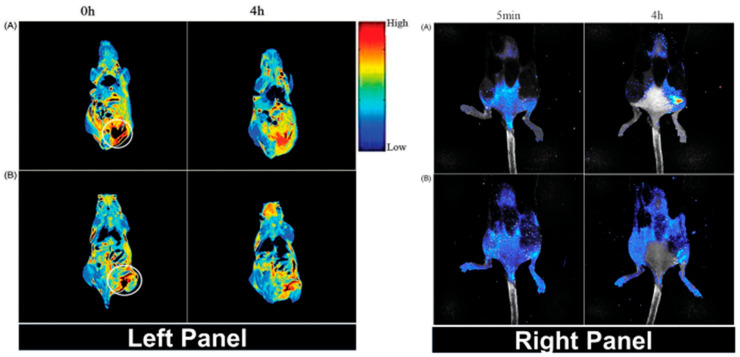
Whole-body T_2_ images in pseudo-color mode (Left Panel) and fluorescent images (Right Panel) in a bone metastasis model at different time intervals of injection of (**A**) cyclic arginine-glycine-aspartate (cRGD)-conjugated magnetic-fluorescent (MF) liposomes and (**B**) unconjugated MF-liposomes. White circles in the left panel represent tumor sites. Reprinted with permission from ref. [44]. Copyright 2015 Taylor & Francis.

**Figure 5 biomedicines-09-00288-f005:**
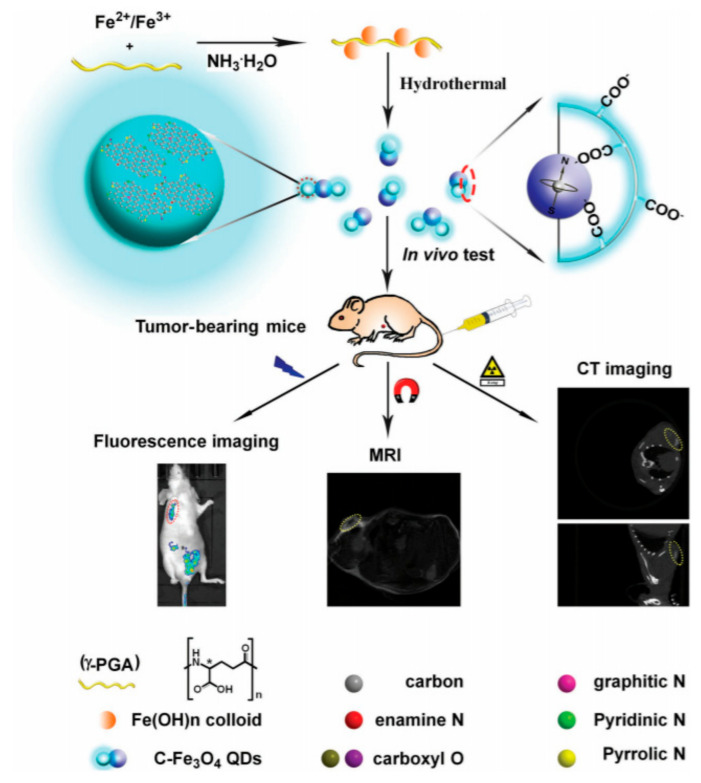
Schematics for the synthesis of C-Fe_3_O_4_ quantum dots (QDs) showing the multimodal imaging application in vivo using HeLa tumor-bearing nude mice. Dotted circles indicate tumor sites. Reprinted with permission from ref. [48]. Copyright 2016 John Wiley and Sons.

**Figure 6 biomedicines-09-00288-f006:**
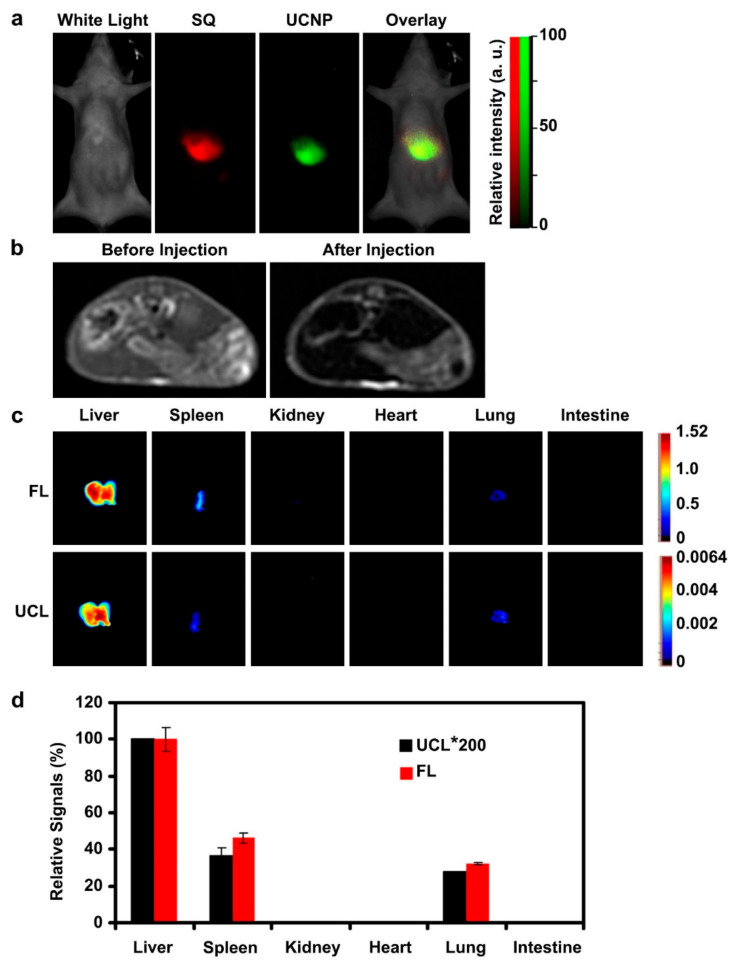
In vivo up-conversion luminescence (UCL)/fluorescent (FL)/MRI images of mice after intravenously injected up-conversion nanoparticles (UCNP)-iron oxide nanoparticles (IONP)-Polymer nanocomposite with Squarine (SQ) dye: (**a**) UCL (green) and FL (red) images (**b**) T_2_-weighted MRI images of mice before and after injection showing darkening effect in the liver. (**c**) Ex-vivo UCL/FL images of major organs post-injection. (**d**) Averaged UCL and FL signals of whole organs relative to the liver. Reprinted with permission from ref. [57]. Copyright 2011 Elsevier.

**Figure 7 biomedicines-09-00288-f007:**
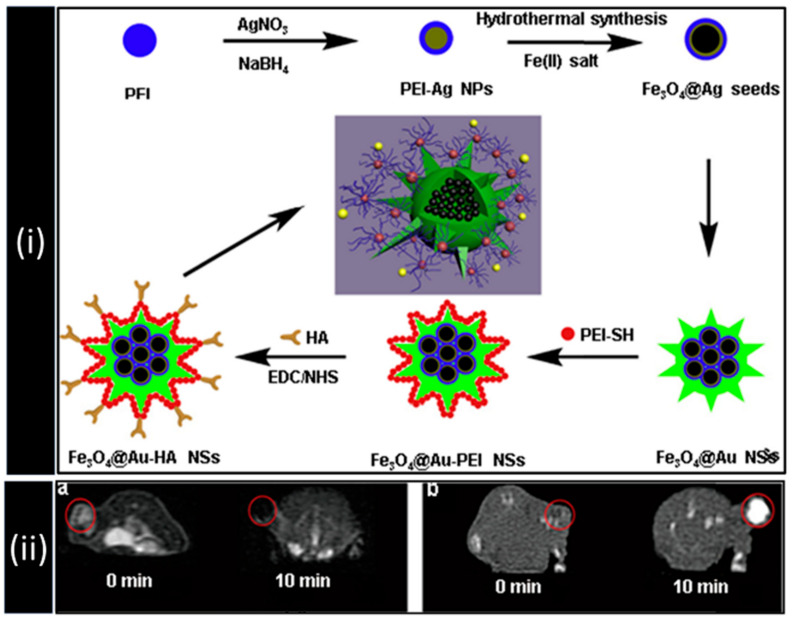
(**i**): Schematic illustration of the synthesis of Fe_3_O_4_@Au- hyaluronic acid (HA) nanostars (NSs). (**ii**): (**a**) T_2_-weighted MRI scan and (**b**) CT images of the tumors in a xenografted tumor model before injection and 10 min post intratumoral injection of nanocomposites. Red circles in Figure 7ii represent selected tumor sites. Reprinted with permission from ref. [69]. Copyright 2015 Elsevier.

**Figure 8 biomedicines-09-00288-f008:**
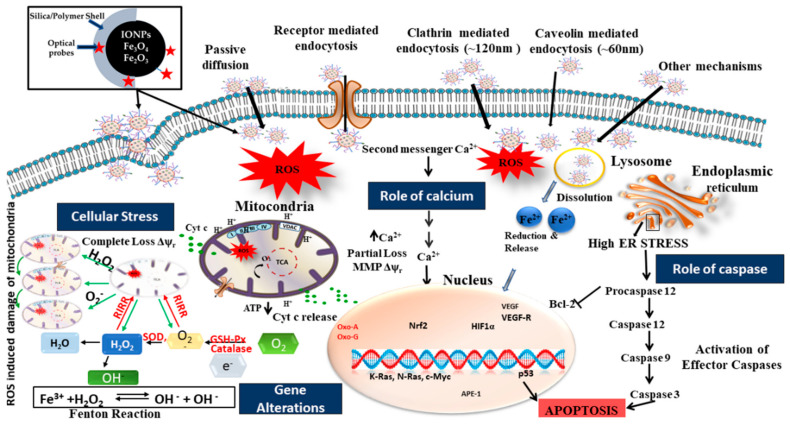
Possible mechanisms of iron oxide-based nanocomposites induced toxicity at the cellular level.

**Figure 9 biomedicines-09-00288-f009:**
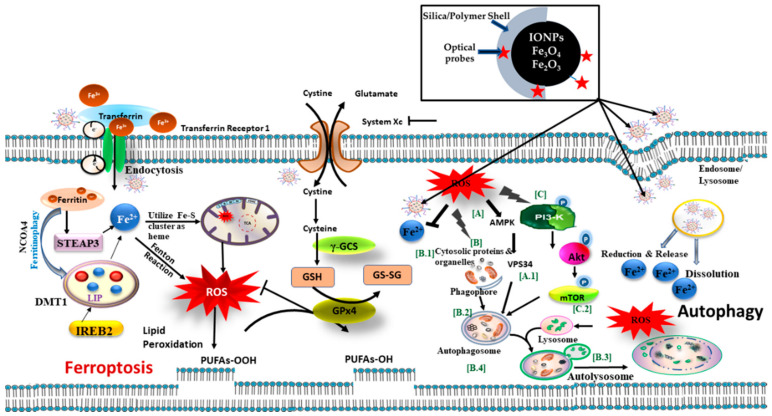
Molecular pathways induced by iron oxide-based nanocomposites.

**Figure 10 biomedicines-09-00288-f010:**
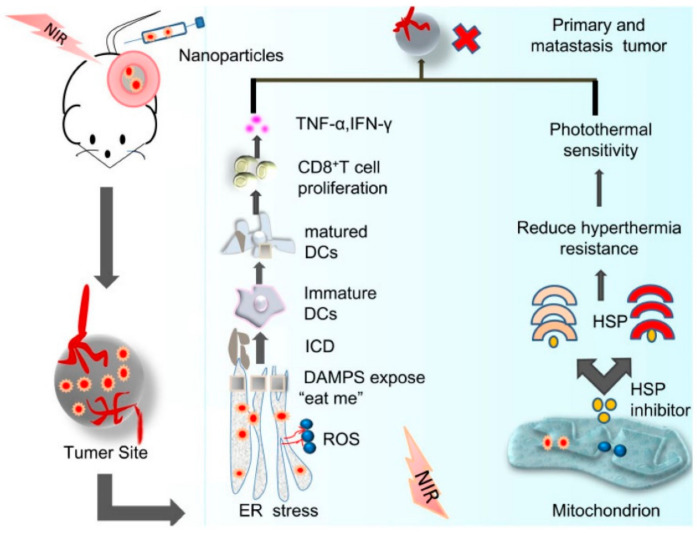
Possible pathways for immunogenic cell death due to the combination of PTT and PDT. Reprinted with permission from ref. [101]. Open Access Creative Common.

**Table 1 biomedicines-09-00288-t001:** Plasmonic nanoparticles—iron oxide nanocomposites for in vivo biomedical application.

Nanoparticles Composition	Synthesis (Magnetic NPs)	Synthesis (Optical NPs)	Synthesis (Nanocomposites)	Magnetization (emu/g)	T_2_ Relaxivity (mM^−1^ s^−1^)	In Vivo Tumor Model	In vivo Application
Fe_3_O_4_-Au-poly(DMA-r-mPEGMA-r-MA)	Thermal decomposition	Thermal decomposition	Emulsion	N/A	245	MCA-TL cells bearing orthotopic hepatoma mice model	CT and MRI dual contrast agents for hepatoma imaging [64]
Magnetic cluster-Au nanorods	High-temperature hydrolysis method	Seed mediated	Microfluidic fabrication (Droplet gelation process)	N/A	r_2_(=1/T_2_) =15.2 mg^−1^s^−1^	HCC orthotopic rat model	Improving the treatment of hepatic malignancies through transcatheter intra-arterial drug delivery system with MRI and CT imaging [79]
IONPs-Au-PEG	Commercial (EMG 304) coated with silica by a gel-sol method	Seed mediated growth of gold shell on IONPs	Nano-shell coated with PEG	3.5 (at 20 kOe)	369	Athymic nude mice bearing orthotropic U87 tumors	To label MSCs to track nanocomposite for MRI and PA imaging [71]
Fe_3_O_4_-Ag @Au-HA nanostars	Hydrothermal	Seed mediated	Modified with PEI and HA	N/A	144.39	Hela tumor-bearing mice	MRI/CT contrasting for tumor imaging and Photothermal imaging mediated therapy under 915 nm laser irradiation (1.2 W/cm^2^) [69]
Fe_2_O_3_-Au-FA	Co-precipitation	Seed mediated	Conjugation of cysteamine-FA	~40	N/A	CT26 bearing colon tumor	PTT under NIR irradiation (808 nm, 1.4 W/cm^2^) for efficient eradication of tumor cells with MRI guidance under MAT [73]
Fe_3_O_4_ @Ag@ Carbon-PEG-FA/Dox	Solvothermal reaction	Solvothermal reaction	Hydrothermal reaction with addition of PEG, FA, and Dox	102	82.1	Hela tumor-bearing mice	MRI/FL for tumor imaging and PTT under NIR irradiation at 808 nm (1.5 W/cm^2^) [74]
PPy@Fe_3_O_4_ /Au	Emulsion polymerization with PVA	Nucleation	Electrostatic adsorption	-	360.8	HeLa cell-bearing nude mice	MR and CT imaging-guided efficient photothermal ablation of tumors [80]

IONPs: iron oxide nanoparticles, Au: Gold, Ag: Silver, DMA: dodecyl methacrylate, mPEGMA: poly(ethylene glycol) methyl ether methacrylate, MA: methacrylic acid, HA: Hyaluronic Acid, FA: Folic Acid, dox: Doxorubicin, CT: Computed tomography, MRI: Magnetic resonance imaging, PA: Photoacoustic, PTT: Photothermal therapy, NIR: Near infra-red, MAT: Magnetically assisted targeting, N/A: Not available.

## Data Availability

Not Applicable.

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
