# Peer review of "Iron Oxide-Based Magneto-Optical Nanocomposites for In Vivo Biomedical Applications"

_biomedicines, 2021, doi:10.3390/biomedicines9030288_

Round 1

Reviewer 1 Report

Paper entitled “Iron oxide-based magneto-optical nanocomposites for in-vivo biomedical applications” meets the necessary standards for publication in this journal. Please check the entire manuscript carefully for eventual typographical errors. Final Conclusion: The paper meets the necessary standards for publication.

Author Response

We highly appreciate to reviewer 1 for time to read and providing positive feedback to our article.  We have carefully proof-read the article and found some typos and spelling mistakes which we have now corrected.

Reviewer 2 Report

The article describes the role of IONPs in the development of nanomedicines. It is a very important subject to the medical field and, in general, to science.  

Here are some comments:

  1. Although the study is very important, there is no indication of how IONPs and the process they are involved in effect from the pH. The pH is an important factor which could effect the nature of IONPs.
  2. Fenton reaction is mentioned in the review; I think the authors have to expand about IONPs in the aspect of Fenton reaction because this reaction is very important in the biologic system.

Author Response

Response to reviwer 2

We highly appreciate to reviewer 2 too for time to read and providing valuable feedback with 5* status in all 5 questions.  Please see our response to the questions reviwer has raised.  We have already instere them in the reviserd version of our manuscript and highlighed in yellow.

The article describes the role of IONPs in the development of nanomedicines. It is a very important subject to the medical field and, in general, to science.  

Here are some comments:

  1. Although the study is very important, there is no indication of how IONPs and the process they are involved in effect from the pH. The pH is an important factor which could effect the nature of IONPs.

Please see the response below which we have added in page 23, lines 883-890 in our revised manuscript

pH can play an important role for controlling the nature (shapes, sizes and zeta potential) of IONPs core during the fabrication [108].  However, in the context of this review article on in vivo biomedical application, pH of the cellular system can also influence the formation of iron ions (Fe2+) which is responsible for Fenton reaction and additional oxidative stress especially in the tutor sites where the pH is mostly acidic.  The variation of shapes and sizes of IONPs core with optical probes may have an influence on the cellular diffusion hence the formation of iron ions (Fe2+) in the tumor sites.

  1. Mahmoudi, M.; Sant, S; Wang, B; Laurent, S; Sen, Superparamagnetic iron oxide nanoparticles (SPIONs): Development, surface modification and applications in chemotherapy Adv. Drug Deliv. Rev. 2011, 63, 24-46.

  1. Fenton reaction is mentioned in the review; I think the authors have to expand about IONPs in the aspect of Fenton reaction because this reaction is very important in the biologic system.

Please see the response below which we have added in page 21, lines 793-795 in our revised manuscript

The co-existence of iron ions (Fe+2) from IONPs and H2O2 in cellular system initiates the Fenton reaction (Fe2+ + H2O2 → Fe3+ + ⋅OH + OH), produces hydroxyl radical (⋅OH), a powerful oxidant creating additional oxidative stress on the tumor cells.